# Superhydrophobic Aluminium Surface to Enhance Corrosion Resistance and Obtain Self-Cleaning and Anti-Icing Ability

**DOI:** 10.3390/molecules27031099

**Published:** 2022-02-07

**Authors:** Peter Rodič, Barbara Kapun, Ingrid Milošev

**Affiliations:** Department of Physical and Organic Chemistry, Jožef Stefan Institute, Jamova cesta 39, 1000 Ljubljana, Slovenia; barbara.kapun@ijs.si (B.K.); ingrid.milosev@ijs.si (I.M.)

**Keywords:** aluminium, superhydrophobic surface, corrosion, self-cleaning, anti-icing

## Abstract

A facile environmentally acceptable surface roughening method using chemical etching in HCl/H_2_O_2_ followed by grafting with n-octyltrimethoxysilane (AS-8) and 1H,1H,2H,2H-perfluorooctyltrimethoxysilane (FAS-8) was studied to fabricate a (super)hydrophobic aluminium surface. The ground aluminium surface after selected etching times (before and after grafting), was characterised using a contact profilometer, optical tensiometer, scanning electron microscope coupled with an energy-dispersive spectroscope and X-ray photoelectron spectroscope to evaluate surface roughness, wettability, surface morphology and composition. The durability of the grafted surface was tested using thermal and UV resistance tests. The corrosion properties were evaluated using potentiodynamic measurements and standard salts spray testing, ASTM B117-19. Finally, the self-cleaning and anti-icing abilities were assessed. The grafted aluminium surface with octyl- or perfluorooctyl silane reflected the highly hydrophobic (AS-8) and superhydrophobic behaviour (FAS-8). Moreover, the different behaviour of the octyl- or perfluorooctyl chain in the silane molecule on modified surface properties was also noticed because durability tests confirmed greater thermal, UV stability and corrosion resistance of FAS-8 compared to AS-8. The aluminium etched for 2 min and grafted with FAS-8 also demonstrated an excellent self-cleaning and anti-icing performance.

## 1. Introduction

Aluminium is one of the most often used metals in the modern engineering industry due to its superior physical and mechanical properties [1,2]. Its corrosion resistance under atmospheric conditions is based on the compact and chemically stable native aluminium passive oxide film formed on its surface. In more corrosive environments, such as chloride-containing solutions, the passive film loses its barrier properties, and the aluminium corrodes [2,3].

There are several surface treatments to enhance corrosion resistance and prolong the life span of aluminium. In the past, chromate conversion coatings have been widely used to inhibit such corrosion events on aluminium and aluminium alloys [4,5,6,7,8]. Nowadays, due to strict environmental regulations worldwide, various methods have been suggested as an alternative, such as cerium [9,10,11,12,13] and zirconium conversion coatings [14], hybrid sol-gel coatings [15,16,17,18,19] and eloxation (anodisation) [20,21,22]. One of the most appropriate novel approaches is a metal surface modification to obtain the superhydrophobic surface [23,24,25,26,27]. Such a modified surface has a water contact angle greater than 150° and a sliding angle of less than 10° [26,28,29,30], and gives highly water-repellent properties, reflected improved anti-corrosive [23,24,25,26,27,31] as well as self-cleaning [31,32,33,34,35,36,37] and anti-icing properties [31,33,37,38,39,40,41]. In the last two years, the superhydrophobic surface has also been attractive for other applications to prevent microorganism growth and virus spread [42,43]. Due to these properties, surface treatment is attractive for many academic and industrial areas. 

Up to now, various artificial superhydrophobic surfaces have been fabricated by tailoring surface topography and using techniques such as electrodeposition [30,44,45], anodic oxidation [46,47], chemical/plasma etching [32,37,48,49,50] and sol-gel processing [51]. However, special equipment or sophisticated control of preparation parameters is required in many preparation processes. Such methods are expensive and time-consuming, and consequently not of interest for low-cost production and application. Therefore, the research work in many research centres is devoted to developing a cheaper, faster, more comfortable and environmentally acceptable method appropriate for the industry [23,26,30].

The superhydrophobic aluminium surface can be achieved following two key factors: the micro-nano scale (hierarchical) structure produced by etching the surface and low surface energy obtained by grafting low-surface energy materials [28]. The well-accepted chemically reactive chemical etching of Al is obtained by copper salts (CuCl_2_ or CuSO_4_) [52] or FeCl_3_ [37] as a result of the displacement reaction of Al and Cu/Fe. Such an etching process may possess many advantages, such as being ultra-fast as well as an environmentally friendly reaction. A well-accepted chemical etching method is also immersion in alkaline (i.e., NaOH [34,49,53], KOH [48]) and acidic solutions (i.e., HCl [36], in combinations of HCl/HNO_3_ solutions [35]). However, recent research confirmed that the etching efficiency in HCl solution could be enhanced with the addition of hydrogen peroxide in a specific ratio (HCl/H_2_O_2_) [33].

After etching, the obtained hierarchical surface structure is superhydrophilic due to the presence of many hydroxyl groups on the metal surface. Therefore, in the next step, it has to be grafted with molecules with a low surface energy. Recently, numerous researchers confirmed that the surface modification could be achieved by surface treatment during immersion ethanol or toluene solutions containing carboxylic acids with a long perfluoroalkyl chain [34,36,49] or silanisation using perfluoroalkyl silanes [31,53]. The latter is chemically bonded on the aluminium surface with strong chemical Al–O–Si bonds, which gives good adhesion to the surface [31,54,55]. However, the superhydrophobicity is also related to the lowered surface energy by introducing alkyl (–CH_2_)*_n_*–CH_3_ and perfluoroalkyl chains (–CF_2_)*_n_*–CF_3_ where *n* presents the chain length. Two important parameters have to be considered: a) the surface energy of the formed film due to the difference of alkyl and perfluoroalkyl chain [56] and b) interactions between chains in the formed film [57]. Such hierarchical structure and formed film with low surface energy allow the air in the formed structure to be trapped, making the surface superhydrophobic [24,25,27]. As a result, the corrosive medium cannot react with the metal surface, and the surface remains protected from corrosion. The formed film must be thermally stable and durable during exposure to UV light [29]. According to the literature, it is expected that a perfluoroalkyl chain compared to an alkyl chain has better thermal [57] and UV stability [58].

In our former studies, the superhydrophobic surfaces in Al were prepared by one-step ultrasound fabrication in NaOH followed by the grafting of alkyl and perfluoroalkyl silanes of different chain lengths [31], and by chemical etching in FeCl_3_ followed by grafting with perfluoro silane [37]. In this study, different parameters during a two-step process (etching and grafting the aluminium surface) were studied. The first improvement is related to the optimisation of the etching process in HCl/H_2_O_2_ solution at various etching times. Second, the comparison of surface treatment performance using alkyl and perfluoroalkyl silanes was characterised. The etched aluminium samples were immersed in an ethanol solution containing n-octyltrimethoxysilane (AS-8) or 1H,1H,2H,2H-perfluorooctyltrimethoxysilane (FAS-8). The surface characterisation was performed using contact profilometry, optical tensiometer, scanning electron microscopy (SEM) coupled with an energy-dispersive X-ray spectroscopy (EDS) and X-ray photoelectron spectroscopy (XPS). The durability tests were performed during thermal and UV light exposure. The corrosion properties were evaluated using potentiodynamic measurements and standard salt spray test ASTM B117-19. Additionally, the self-cleaning and anti-icing properties were monitored.

## 2. Experimental Details

### 2.1. Metal Substrates and Chemicals

Commercially available aluminium specimens with a size of 2 cm × 2 cm × 1 cm were cut from the aluminium foil (99.0%, distributed by GoodFellow GmbH, Hamburg, Germany, Europe). The surfaces were ground with a Struers LaboSystem LaboPol-20 machine using 1000, 2000 and 4200 SiC abrasive papers (supplied by Struers ApS, Ballerup, Denmark, Europe) in sequences in the presence of tap water. Then the specimens were thoroughly rinsed with distilled water and pure ethanol in an ultrasonic bath (Elmasonic S10H, Lab Logistics Group–LLG, Meckenheim, Germany, Europe) to remove all grinding residuals and other organic substances.

The two-step fabrication of the treated aluminium surface included chemical etching in a hydrochloric acid/hydrogen peroxide (HCl/H_2_O_2_) solution at selected times (from 30 s to 3 min with 30 s interval) followed by grafting in the ethanol solution containing n-octyltrimethoxysilane or 1H,1H,2H,2H-perfluorooctyltrimethoxysilane, Figure 1. 

The etchant solution was prepared by mixing H_2_O (Milli-Q Direct water with the resistivity of 18.2 MΩ cm at 25 °C, Millipore, Billerica, MA), 37 wt. % HCl (CAS No. 7647-01-0, ACS reagent, Sigma-Aldrich, St. Louis, Missouri, USA), and 30 wt. % H_2_O_2_ CAS No. 7722-84-1, ACS reagent, Sigma-Aldrich, St. Louis, Missouri, USA) in the volume ratio of 10:2:3. The aluminium samples were immersed into a freshly prepared etching solution in the glass tempering beaker to control the etching solution temperature between 25 and 40 °C. The ground aluminium surface was faced up, and the system was opened to air. After etching for various selected times (ranging from 0.5 to 3 min), the specimens were immediately thoroughly rinsed with distilled water, cleaned ultrasonically in pure ethanol and dried with a stream of compressed nitrogen.

In the next step, the grafting of the etched samples was performed by immersion in freshly prepared 1 wt. % ethanol solution containing n-octyltrimethoxysilane (AS-8, CAS No. 3069-40-7) or 1H,1H,2H,2H-perfluorooctyltrimethoxysilane (FAS-8, CAS No. 85857-16-5), both distributed from Fluorochem, Hadfield, Derbyshire, United Kingdom, Europe and used without any purification. The samples were placed in a beaker with the etched surface facing up. This process was performed at an ambient temperature. After 30 min, the samples were taken out, thoroughly rinsed with distilled water and ethanol, and dried with a stream of compressed nitrogen.

### 2.2. Surface Characterisation 

#### 2.2.1. Weight Loss Test

The weight loss test was performed to determine the rate of the etching process after various selected etching times. The process was quantitively evaluated by calculating the weight loss given in percentage (W (%)) between initial sample weight (W_initial_) and etched samples weight (W_etched_) of a clean aluminium following Equation (1):(1)$$W\left(\%\right)=\frac{W_{\mathrm{initial}}-W_{\mathrm{etched}}}{W_{\mathrm{initial}\;}}\times 100$$

The evaluation was performed on five parallel samples to achieve good accuracy. The obtained values are given as average values ± standard deviation.

#### 2.2.2. Surface Topography

Surface 3D topography of the ground and etched aluminium after selected etching times was evaluated on three randomly chosen spots to assess the changes in the rough surfaces. Measurements were performed employing a stylus contact profilometer, model Bruker Stylus profilometer, DektakXT, using a 2 μm tip and a soft-touch mode with force 1 mN. The measurements were performed on 1 mm × 1 mm large area, with the vertical analysis range 65.5 µm and the vertical resolution 0.167 µm/point. The obtained data were analysed using TalyMap Gold 6.2 software. From the obtained 3D mapping, their corresponding surface roughness (*S*_a_) was calculated according to the standard ISO 25178-1:2016. The values are given as mean values ± standard deviations.

#### 2.2.3. Surface Wettability

Water contact angle (WCA) and diiodomethane CH_2_I_2_ (99 %, CAS Number 75-11-6, SigmaAldrich, St. Louis, MI, USA) contact angle (DCA) measurements were performed at an ambient temperature by the static sessile-drop method on a Krüss FM40 EasyDrop contact-angle measuring system. The polar surface energy *γ*_l_ for the test liquids are γ_l_ = 72.8 mN/m for water and γ_l_ = 50.8 mN/m for diiodomethane [59]. A small liquid droplet (6 μL) was formed on the end of the disposable needle (stainless steel, with a PP luer-lock connector, with a length of 38 mm and diameter of 0.5 mm; Krüss NE 44), which was then carefully deposited onto the ground and treated aluminium surface. Digital images of the droplet silhouette were captured with a high-quality camera, followed by the contact angle determination by numerically fitting the droplet image. The values reported herein were the average of at least five measurements on various parts of each sample. 

#### 2.2.4. Durability Tests

The wettability of a grafted surface deteriorates due to the degradation of hydrophobic organic molecules at high temperatures. The thermal stability of the modified aluminium surface was checked by annealing for one hour the treated aluminium at various temperatures from 80 to 200 °C at an interval of 10 °C. The thermal stability of the formed film was evaluated by measuring the water contact angle (WCA) after the sample was cooled down to an ambient temperature.

Additionally, the durability of the grafted surface during UV radiation was also studied. The samples were exposed to ultraviolet light using UV Ultraviolet LED Flashlight of wavelength 275 nm and irradiance 2.5 mW/cm^2^ from 3 cm distance exposure for one week. The WCA changes were evaluated to evaluate the effect on wettability at the tested time interval.

#### 2.2.5. SEM/EDS Characterisation

A field-emission scanning electron microscope JEOL JSM 7600 F equipped with an energy-dispersive X-ray spectrometer (EDS) Inca 400, Oxford Instruments, were used to analyse the morphology and composition of etched aluminium in HCl/H_2_O_2_ and grafted with AS-8 and FAS-8. Before analysis, samples were sputter-coated with a thin layer of gold (Au) using sputter coater BAL-TEC SCD 005 to provide an electrically conductive thin film. The FE-SEM imaging was performed using a secondary electron detector (SEI mode) and backscattering electrons for composition (COMPO mode) of the specimens at 5 and 10 kV voltage. The EDS analyses were performed at 10 kV in a point analysis mode. The data were normalised and given as a weight percentage, wt. %. The Au was excluded from the quantitative analysis. However, part of the carbon peak (i.e., 2 wt. %) can be related to artificial carbon. 

#### 2.2.6. XPS Characterisation

The chemical composition of the surface was analysed using X-ray Photoelectron Spectroscopy (XPS) using instrument XPS spectrometer PHI TFA XPS, Physical Electronics USA, equipped with aluminium and magnesium monochromatised radiation. An XPS survey and high-resolution analysis were carried out using Al-K_α_ radiation (1486.92 eV). The measurements were done in a constant analyser energy mode with 187.9 eV pass energy for survey spectra and 39.35 eV pass energy for high-resolution spectra. Photoelectrons were collected from an angle of 45° relative to the sample surface. Charge referencing was done by setting the low binding energy C 1s photopeak at 284.8 eV. The elemental composition was determined from the high-resolution peak analysis. 

#### 2.2.7. Corrosion Testing

Linear polarisation, potentiodynamic measurements and electrochemical impedance spectroscopy (EIS) measurements were performed using an Autolab Potentiostat/Galvanostat, Model PGSTAT12, controlled by the Nova 2.1 software. Testing was performed with the 250 mL electrochemical cell with a configuration of three electrodes: working, counter and reference electrodes. As a working electrode, the ground, etched and grafted aluminium were tested, where a surface of 1 cm^2^ was exposed to the 0.1 M NaCl solution, used as a corrosive medium. A graphite rod was used as the counter electrode and silver/silver chloride Ag/AgCl (3 M KCl, 0.210 V) as a reference electrode. All tests were performed in triplicate at the ambient temperature (23 ± 2 °C). For the potentiodynamic polarisation measurements, the potential was varied from a value −250 mV lower than the open circuit potential (*E*_oc_) up to a few hundred mV above it, till the current density increased rapidly, reaching the pitting corrosion (*E*_pit_). The potential scan rate was 1 mV/s. The representative curves are plotted in Figures, and corrosion current density, *j*_corr_, corrosion potential *E*_corr_ and pitting potentials *E*_pit_ were evaluated from the curves. Additionally, the differences between *E*_pit_ and *E*_corr_ (Δ*E*) were also calculated using Equation (2): Δ*E* = *E*_pit_ − *E*_corr_
(2)

Δ*E* denotes the span of the inhibited region. 

All obtained electrochemical data are tabulated and presented as mean value ± standard deviation.

Additionally, an accelerated corrosion test was performed using salt spray corrosion testing. The tests were performed on 2 cm × 4 cm ground and AS-8 and FAS-10 treated aluminium specimens. The salt spray chamber ASCOTT, Staffs, Great Britain, with 0.17 m^3^ capacity, was operated according to the standard ASTM B117-19. The pH of NaCl solution (50 ± 1% g/L) was set between 6.0 and 6.5 at room temperature to give values of pH between 6.5 and 7.2 after heating the solution to 35 °C. pH was adjusted with 0.1 M NaOH or HCl solutions. The device for spraying the salt solution comprised a supply of clean air of controlled pressure and humidity, a reservoir to contain the solution to be sprayed and a single sprayer. The compressed air was passed through a filter to remove all traces of oil or solid matter. The temperature in the salt spray chamber was set to 35 °C ± 2 °C. The corners of the specimens were protected with waterproof tape. The samples were placed in a holder so that the sample’s surface was tilted at an angle of 30°. 

The volume of the sprayed solution was monitored with standardised collecting devices: a glass funnel (a diameter of 100 mm/having a collecting area of ~80 cm^2^) with the stems inserted into graduated glass cylinders. The collecting device was placed in the cabinet zone where the test specimens were placed. The sprayed volume of the salt solution was set to ~1 mL/hour. After 12 h, 1 week and 2 weeks, the samples were taken from the chamber, thoroughly rinsed with distilled water and dried with compressed nitrogen. Then the sample surface appearances were photographed and compared.

#### 2.2.8. Self-Cleaning

The self-cleaning experiments were carried out by the following procedures: the aluminium sample on a horizontal stage (tilted at 2 degrees). The solid pollution was simulated by covering the aluminium surface with a layer of graphite powdered multiwalled nanotube (carbon > 95 %, length 1–10°µm, PlasmaChem GmbH, Berlin, Germany, Europe) to simulate chalk powder. Then, water droplets of a few µL (with the added blue dye) were dropped from 2 cm high on the surface. The flow of chalk powder, along with water droplets, was recorded to evaluate the dust removal. 

#### 2.2.9. Anti-Icing

The anti-icing properties were studied on non-treated and treated aluminium under overcooled conditions, including the water-dripping test and the freezing delay time test. The test was performed with water droplets of 8 µL, which were gently placed on the ground and AS-8 and FAS-8, horizontally positioned Al samples. The specimens were put into the freezer (–15 °C). After 1 h, the specimens were taken out and left at ambient temperature. The melting process was evaluated by recording the melting times on non-treated and treated surfaces using a digital camera and thermal Fluke Ti55FT infrared camera (Everett, WA, USA). 

## 3. Results and Discussion

### 3.1. Weight Loss and Surface Roughness

The two-step process of the aluminium surface treatment (1st step—chemical etching, followed by 2nd step—grafting) is schematically presented in Figure 1. The etching of aluminium in HCl/H_2_O_2_ is a vigorous reaction, which can be observed as a gas release in bubbles. This process is exothermic; therefore, the increase of etching solution temperature was noticed, but due to the glass tempering beaker, the etching solution temperature was controlled not to exceed 40 °C. Prepared mixture HCl/H_2_O_2_ solution has pH~1, under which conditions the aluminium oxide passive layer is not resistant, and aluminium reacts vigorously [2]. The summary of the chemical process on the aluminium surface during etching is presented with Equations (3)–(5). Aluminium reacts with acid (HCl), forming AlCl_3_. The side product of this reaction is H_2_. At the same time, formed AlCl_3_ spontaneously reacts with hydrogen peroxide forming the aluminium hydroxide, Al(OH)_3_. The chlorine remains dissolved in the solution due to its high solubility in water.
2Al_(s)_ + 6HCl_(aq)_ → 2AlCl_3(s)_ + 3H_2(g)_
(3)
2AlCl_3(s)_ + 3H_2_O_2(aq)_ → 2Al(OH)_3(s)_ + 3Cl_2(aq)_(4)
Cl_2(aq)_ + H_2_O_(l)_ → HCl_(aq)_ + HClO_(aq)_(5)

The etching rate of aluminium in the HCl/H_2_O_2_ mixture was evaluated from the slope of the curve obtained from weight loss calculations after selected times, Figure 2a. 

The aluminium was etched for selected immersion times ranging from 0.5 to 3 min. We noticed nonlinear behaviour reflecting different etching rates. It was slower at the beginning (first 1 min of etching), where the weight loss was above 0.4%/min. This confirmed that the naturally formed aluminium oxide film is more resistant in such an acidic solution than the aluminium underneath. The process took place mainly locally, creating tiny pits in the oxide film. Later (between 1 and 2 min), the passive film was dissolved, and the etching rate increased due to the formation of deep and wide holes in the aluminium (0.5%/min). After a particular time (between 2 and 3 min), the etching rate became almost linear (1.2%/min) with uniform etching of the entire surface. After a longer etching time (3 min), the sample weight loss was above 2.6%, making further evaluation needless.

The surface topography at selected etching times was estimated using a contact profilometer to compare the surface topography changes, surface roughness (*S*_a_) and linear profile of ground aluminium before and after etching in HCl/H_2_O_2_ at selected etching times. The ground aluminium surface was flat and smooth, but due to the softness of the aluminium, some small scribes remained after the grinding process, Figure 3(a1,a2). 

Moreover, some structural defects remain despite the grinding process with SiC papers. 3D images provided immediate topography changes on the etched aluminium surfaces, Figure 2b and Figure 3b–d. After 1 min, there were slight changes because the etching process was slow due to the presence of the passive film (*S*_a_ = 0.68 µm). The increasing surface roughness agrees with the mass loss during the etching process, Figure 2a. There were mainly tiny randomly spread pits formed in the passive aluminium film, Figure 3(b1), which are better seen in the line profile (etched for 1 min), Figure 3(b2). After a prolonged etching time, the passive layer was removed, and the sample roughness was changing almost linearly with etching time between 1 and 2 min up to *S*_a_ = 5.51 μm, Figure 2b. After 2 min, the surface topography rapidly shows a significantly rougher surface due to broader and deeper pits, Figure 3(c1). The depth of the pits is up to 30 µm, Figure 3(c2). After a longer etching time (3 min), the pits were enhanced into larger (*S*_a_ increased to 7.2 m, Figure 2b and Figure 3(d1)) and even deeper holes (up to 40 µm), Figure 3(d2).

The sample weight and surface roughness changes were also evaluated after grafting the specimens with AS-8 and FAS-8. Due to the low thickness of the formed silane film on the etched aluminium surface, the changes in sample weight and surface roughness cannot be detected with these techniques (data are not presented additionally in the text). However, it can be concluded that the changes of the specimen weight and its increased surface roughness were obtained during the first step of surface treatment (etching process).

### 3.2. Surface Wetting Behaviour

The wettability of the ground and AS-8- and FAS-8-treated aluminium surfaces was evaluated by measuring the water (WCA) and diiodomethane (DCA) static contact angles, Figure 4.

The ground aluminium has a low WCA, near 58°, and water as polar liquid (*γ*_l_ = 72.8 mN/m) wets its surface. Similar wettability reflects ground aluminium surfaces (without etching, i.e., etching time was 0 min) grafted with AS-8 because WCA did not exceed 60°, which confirmed that the treated surface with AS-8 did not change the WCA. A similar WCA was also noticed for 1 min etched aluminium grafted with AS-8. This confirms that the etching step was inefficient in removing the oxide film and forming a hierarchical structure to increase the WCA after grafting with AS-8. On the other hand, the WCA increased significantly up to 132° for aluminium etched for 2 or 3 min. Such increased behaviour of WCA confirmed the importance of surface pre-treatment (etching) before grafting the aluminium surface with AS-8.

On the other hand, the DCA as non-polar liquid are very small (only a few degrees) because non-polar diiodomethane (*γ*_l_ = 50.8 mN/m) almost entirely wets the surface. There was only a slight difference between grafting the ground and etched surface with AS-8. The DCA slightly increased with etching time, but the maximum DCA was only 50 °. This agrees with the non-polar behaviour of C–H groups in the chain of AS-8 molecules, which does not reflect oleophobicity.

On the other hand, despite the similar roughness of etched aluminium surfaces and grafting mechanism, there was a significant improvement in wetting the surface for all grafted aluminium surfaces with FAS-8. Such behaviour reflects the vital difference in the surface energy of alkyl and perfluoroalkyl groups, meaning that the surface energy of FAS-8 is lower than that of AS-8. The WCA increased with etching time and the superhydrophobic surface (WCA > 150°) was obtained for 2 min etched aluminium surfaces grafted by FAS-8. Extended etching time (to 3 min) increased the WCA up to 156° and improved reproducibility. The grafting of aluminium with FAS-8 significantly affects the DCA, which is much greater than DCA for aluminium grafted with AS-8. The DCA for Al samples etched for two and three minutes was above 130°, reflecting more polar CF_3_ groups in the chain of the FAS-8 molecule.

These results confirmed that despite the equal roughening method, it was crucial to choose silane molecules with low surface energy to obtain the surface with (super)hydrophobic and oleophobic properties.

### 3.3. Durability Tests

The durability of superhydrophobic surfaces is an essential obstacle to practical applications. Therefore, the prepared (super)hydrophobic surfaces were evaluated through thermal and UV durability tests according to the standards [29], where WCAs were measured after selected increased temperature and UV exposure intervals.

The thermal stability was performed for 2 min etched and grafted aluminium with AS-8 and FAS-8, which showed high hydrophobicity (WCA = 127.5°) or superhydrophobicity (150.2°). However, after heating the samples for one hour at the selected temperature, the difference in the thermal stability of the grafted surface with octyl and perfluorooctyl chain was noticed. The alkyl chain decomposed earlier (around 150 °C), consequently the WCA decreases with time and water droplets wet the surface and adhere to the rough surface.

On the other hand, FAS-8-treated aluminium surface remained constant (above 140°) up to 170 °C, Figure 5a.

Above this temperature, the hydrophobicity was also lost due to the decomposition of perfluorooctyltrimethoxysilane. The obtained data confirmed better thermal stability of FAS-8 molecules on grafted aluminium compared to AS-8.

The UV resistance was examined for similar specimens (2 min etched aluminium and grafted with AS-8 and FAS-8) by exposing the grafted surface to UV light for selected illumination times, Figure 5b. The WCA on an AS-8-treated aluminium surface decreased with exposure time. The decrease of WCA is observed for the alkyl chain, which is broken under UV irradiation leading to the decomposition of the molecules. Consequently, part of the formed film on the etched aluminium surface is destroyed. As a result, WCA decreased and water droplets wet the surface and adhered to the rough surface. After more prolonged exposure, the wettability decreased further and the standard deviation of the WCA increased.

On the other hand, the aluminium treated with FAS-8 still exhibited superhydrophobicity with a contact angle above 144° even after 7 days of exposure under UV light, which suggested an excellent UV radiation stability. Only a slight increase in the standard deviation with exposure time was noted. These results are in accordance with the high bond energy (~5 eV) of the C–F bonds in the FAS-8 molecule compared to C–H (~3.8 eV).

The obtained data revealed that the durability of the treated aluminium surface was affected by the type of silane during grafting. Perfluorooctyl silane has much greater thermal stability and resistance to UV light; consequently, such a treated aluminium surface has better durability.

### 3.4. SEM/EDS Surface Characterisation

The surface morphologies of 2 min etched and grafted aluminium with AS-8 and FAS-8 were characterised by SEM microscopy, using secondary electron imaging (SEI mode) at low and backscattered-electron imaging (COMPO mode) at high magnifications, Figure 6.

Both surface appearances of AS-8- and FAS-8-treated aluminium surfaces are very similar. The treated aluminium surface has many spherically patterned Al_2_O_3_ surfaces, with an average pattern size of only a few micrometres. This type of morphology increases the surface roughness of aluminium, Figure 6(a1,b1). Furthermore, it can be noticed that they are composed of hierarchical micro/nano structures, Figure 6(a1,b1). Such structure and grafting allow air to be trapped within, which is essential for obtaining (super)hydrophobic properties. The thickness of the formed film on the etched surface is nano-metric; therefore, the adsorbed molecules cannot be seen in the SEI or COMPO images.

The presence of AS-8 and FAS-8 molecules on the treated aluminium surface was confirmed using point analysis using EDS at numbered spots, spectra in Figure 6(a2,a3,b2,b3). The analysis of micro/nano structures confirmed that the surface consists of Al, O, Si and C, Table 1. 

This demonstrates that the formed AS-8 film can be detected on the surface despite the low thickness of the formed film. The Al and O are related to the Al/Al_2_O_3_ on the surface; on the other hand, the Si and O are attributed to AS-8 molecules bonded on the aluminium surface. The difference in the concentration of Si was observed. A small amount of Si was noticed in the region with a high amount of Al (spots 1,2), but a greater amount can be seen in the region, where Fe has been detected in the spots spread in the aluminium (spot 3). Fe is present as an impurity in the as-received aluminium, which remains in its structure during the etching process. However, it is important to point out that no chloride ions were detected on the treated surface (that can remain after etching). 

A similar amount of Si was also obtained for the aluminium grafted in the FAS-8 solution, Table 1. The main difference in the spectra is the presence of F atoms in the EDS spectra (spots 4-6). These spectra confirmed that the FAS-8 molecules were bonded on the aluminium surface. However, there are some differences in the concentrations of F in different analysed spots. Again, a higher number of FAS-8 molecules can be detected in spots where Fe is present (spot 6). 

### 3.5. XPS Characterisation

The AS-8 and FAS-8 film XPS spectra were acquired to investigate whether the molecules successfully grafted onto the aluminium surface and the changes in the surface chemical composition. The XPS survey scans of AS-8- and FAS-8-treated aluminium surfaces are presented in Figure 7a, and the atomic concentrations of elements in each sample are inserted at the top right corner of the image. 

The Fe was present as impurity in the aluminium, therefore it was excluded from further analysis. The XPS spectra showed bands for aluminium Al 2p, O 1s, C 1s and Si 2p as a substrate for AS-8 ((CH_3_O)_3_–Si–(CH_2_)_7_CH_3_) grafted surface and the presence of Al 2p, O 1s, C 1s, Si 2p and F 1s for FAS-8 (CH_3_O)_3_–Si–(CH_2_)_2_(CF_2_)_5_CF_3_) grafted surface (see molecule structures in Figure 1). The atomic composition of AS-8 and FAS-8 spectra confirmed a great amount of F, O, C, and a slight amount of Al and Si, see inserted data in Figure 7a. These results revealed that the (perfluoro)alkylsilane molecules are bonded onto the etched aluminium surface. It was observed that the C content in the composite film surface was different due to different C–H and C–F compositions (greater in AS-8 grafted surface). The great mass concentration of F element in the XPS spectra was ascribed to the long perfluoroalkyl chain of FAS.

In the next step, the high-resolution XPS spectra of AS-8 and FAS-8 grafted surfaces were analysed and compared to further investigate the surface coverage and chemical bonding. The Al 2p and O 1s spectra of etched aluminium confirmed the presence of aluminium oxide/hydroxide, Figure 7b,c. Spectra related the presence of AlOOH (74.7 eV, 531.7 eV) and passivation during etching in a strongly acidic medium (HCl/H_2_O_2_), exposure to air in the atmosphere and grafting process, Al_2_O_3_ (72.5 eV, 530.7 eV) [60,61]. The typical C 1s XPS peak appeared for the AS-8-treated surface and several peaks for the FAS-8-treated surface, Figure 7c. The primary C 1s signal was centred at 284.8 eV, which was assigned to the combined contribution of C–C together with C–Si and C–H after grafting in AS-8 and FAS-8 solutions. The XPS spectra for FAS-8 can be further resolved into seven components, namely, –CF_3_ (293.8 eV), –CF_2_ (292.0 eV), –CF-CF_x_ (291.0 eV), C–F (289.8 eV), –C–CF_x_ (285.7 eV), –C–C (284.8 eV) and –C–Si (283.8 eV). These characteristic bands of the perfluoroalkyl groups (CF_2_ and CF_3_) confirmed the presence of FAS-8 molecules at the surface. The peak assignments agreed well with previously reported values [37]. The CF_3_ and CF_2_ concentrations present in the C 1s spectrum of the FAS-10-treated Al correlate with the molecular structure. The high concentration of CF_3_ and CF_2_ on the surface indicates that the molecules were orientated towards the Si–O bond on the surface, forming Si–O–Al, while the (CH_2_)_2_(CF_2_)_7_CF_3_ tail comprised the outermost surface film. Such a surface composition (orientation) of AS-8 and FAS-8 correlates with the obtained reduced wettability.

The Si 2p peaks comprised three components (Figure 7e) with binding energy at about 101.3 eV for Si–O–C, 102.5 eV for Si–C associated to AS-8 and FAS molecule, confirming the formation of the covalent bond between aluminium oxide/hydroxide and AS-8 and FAS-8 and at 103.5 eV (75.5 eV in Al 2p spectra, Figure 7b) associated with the Si–O–Al species formed during grafting. The presence of the FAS-8 molecule on treated aluminium can also be confirmed in F 1s spectra with a broad peak between 686–690 eV, Figure 7f. This peak is related to C–F (–CF_2_, –CF_3_) covalent bond in the FAS-8 molecule (Figure 1). 

### 3.6. Corrosion Resistance

#### 3.6.1. Potentiodynamic Polarisation Measurements

The corrosion properties of ground, etched and treated aluminium were evaluated using potentiodynamic polarisation measurements, Figure 8. 

Obtained electrochemical parameters are given in Table 2. 

Ground aluminium has *j*_corr_ of 0.51 µA/cm^2^ and Δ*E* of 10 mV which reflect low corrosion resistance in tested medium. Such behaviour is due to the presence of Cl^−^ ions which cause corrosion of the native Al oxide surface film. During etching, the native oxide film was removed, and the surface area was largely increased, Figure 3. After etching, the air exposure results in a newly formed oxide that exhibited larger *j*_corr_ compared to ground aluminium.

Smaller *j*_corr_ of the treated samples compared to that of etched Al substrate confirm enhanced corrosion resistance. The *j*_corr_ of the AS-8 grafted sample of 1.72 µA/cm^2^ was somewhat smaller than that of etched sample and *E*_corr_ was shifted to the positive direction. However, the pitting process, reflected in an abrupt increase in current density, began immediately following *E*_corr_ resulting in Δ*E* of 10 mV. 

The *j*_corr_ of FAS-8 grafted aluminium was further reduced (0.03 µA/cm^2^), for almost two orders of magnitude compared to the etched, and *E*_corr_ shifted more positive value (–0.60 V). Cathodic current density was also significantly reduced. Following *E*_corr_, a passive region was established extending up to −0.54 V, resulting in Δ*E* of 60 mV. A positive shift of *E*_corr_ was reported previously [31,37] and can be attributed to the improvement of the corrosion resistance due to the formation of the silane film on Al surface as well as the effect of superhydrophobicity [24,27]. Based on these claims, it can be assumed that the corrosion protection is strongly correlated with WCA and increases in the same order as the WCA, i.e., ground Al < AS-8 < FAS-8. It has been generally accepted that the trapped and confined air pockets of the superhydrophobic surfaces behave as a dielectric, which inhibits the electron transfer between the metallic substrate and the electrolyte [23,24]. Hence, the enhanced corrosion properties of AS-8- and FAS-8-treated surfaces are the result of protective silane film with low surface energy as well as hydrophobicity of the surface. Due to low corrosion resistance of etched sample, the following tests were performed only on ground and AS-8- and FAS-8-treated aluminium surfaces. 

#### 3.6.2. Salt Spray Testing 

The corrosion resistance in a severe environment for the ground and treated aluminium with AS-8 and FAS-8 was evaluated at the different periods using neutral salt-spray testing, Figure 9. 

The ground aluminium exhibited low corrosion resistance in the salt environment already after 12 h because its surface was severely corroded, with many pitting sites. Most of the area was coated with corrosion products. The amount and the area of corrosion products increased with exposure time (after 1 and 2 weeks). 

Enhanced corrosion resistance was observed for the aluminium treated with AS-8 film. The grafted AS-8 molecules prevented wetting the surface with corrosive medium, consequently the reaction between corrosive ions and aluminium surface. However, the surface was not completely protected because pitting corrosion was observed after 1 week, Figure 9. More corrosion products (seen as slight grey coloured surface) were seen after extended exposure time (2 weeks), but their amount was still much lower than on the ground surface, confirming improved aluminium corrosion protection. 

Much better corrosion protection was observed for aluminium treated with FAS-8, Figure 9. The superhydrophobic surface prevented wetting and blocked the contact of aluminium surface and corrosion medium. No pitting or any corrosion products were seen even after 2 weeks of exposure. This agrees with the electrochemical data presented in the previous section.

In short, the AS-8 and, especially, FAS-8 grafted aluminium exhibited enhanced corrosion protection compared to ground Al up to 2 weeks of exposure in the neutral salt spray chamber, which confirmed good durability under tested conditions.

### 3.7. Self-Cleaning Performance

Self-cleaning ability was studied on the ground, and 2 min etched aluminium grafted with AS-8 and FAS-8. Carbon particles were spread on Al surface as artificial contaminants and then water droplets containing blue dye were dropped. The experiment results before and after the test are shown in Figure 10a,b (Video available online, Appendix A). 

The carbon particles remained on the ground surface after testing, meaning that the surface did not pose self-cleaning performance, Figure 10b. The carbon particles were wetted and remained adhered to the ground aluminium surface. In contrast, aluminium samples treated with AS-8 and FAS-8 retained their (super)hydrophobicity, Figure 10b. The water droplet on the treated surface has a nearly spherical shape without wetting the surface since it allows water droplets to remove the dirt along the rolling path. After the water pouring process, the water remained on the aluminium surface treated with AS-8; on the other hand, a clean surface remained on aluminium treated with FAS-8. This test confirmed that the superhydrophobic surface with a high-water contact angle and a low sliding angle (samples were tilted only for 2°) of aluminium treated with FAS-8 is essential for obtaining the self-cleaning surface. 

This can be explained due to and formed micro/nano hierarchical structure and grafting of the surface with low free energy molecules, Figure 10c. Such a surface poses low adhesion between the carbon particles and the treated surface. At the same time, the aluminium surface cannot be stained. 

### 3.8. Anti-Icing Tests

The melting (de-icing) process of water dropped onto the ground and treated aluminium surface with AS-8 and FAS-8, recorded by the digital and thermal camera. Results are shown in Figure 11.

The droplets on the ground aluminium surface started to melt after 1 min near the edges, and ice remained only in the centre of larger droplets. After 3 min, the droplets were completely melted. Frozen droplets on AS-8- and FAS-8-treated aluminium melted much slower, Figure 11. After 1 min, the droplets on the AS-8-treated surface remained frozen, and slow melting near the edges can be noted after 3 min. After 6 min on AS-8-treated Al, the drop was melted, which is double the melting time of 3 min compared to ground Al. An even slower melting process was noticed on aluminium treated with FAS-8, where the ice in the centre was present after 8 min; drops were completely melted after 10 min, Figure 11.

The obtained results indicated that the melting (regarded as the opposite way of freezing) process of water droplets on the prepared (super)hydrophobic surface was greatly retarded, demonstrating the film’s superb anti-icing property. Such melting delay can be explained by thermodynamics. The dominant factor is the surface properties. The droplet on the ground surface was in direct contact with aluminium allowing the heat to be directly transferred from the aluminium surface. On the other hand, the droplet on the cold surface gains heat from the air in the forms of heat conduction and thermal radiation and absorbs heat by heat conduction [37]. Thus, droplets suspended on (super)hydrophobic surfaces (treated aluminium) have a longer melting delay time compared to ground surfaces. Therefore, the anti-icing effect can be expressed as a function of the contact angle between droplets and the surface. The hydrophilic surface was more susceptible to de-freeze than the hydrophobic surface. These results demonstrate the significant applicable probability of a superhydrophobic surface in preventing ice formation.

## 4. Conclusions

A (super)hydrophobic aluminium surface has been fabricated by a facile two-step processing consisting of chemical etching in HCl/H_2_O_2_ solution followed by grafting (silanisation) using n-octyltrimethoxysilane (AS-8) or 1H,1H,2H,2H-perfluorooctyltrimethoxysilane (FAS-8). 

During etching, a hierarchically rough surface was formed; with the grafting step, silane molecules react with an aluminium surface and form a nanometres-thin film. Grafting with AS-8 increases the WCA to 132° and FAS-8 to 156°. Better thermal and UV durability was achieved for aluminium treated with FAS-8. With an optimal etching time of 2 min and immersion for 30 min in FAS-8 solution, the superhydrophobic aluminium surface (WCA > 150°).

SEM/EDS analyses confirmed the presence of the formed film on the surface. Higher concentrations can be detected on impurities present in the aluminium. XPS analyses confirmed chemically bonded AS-8 and FAS-8 on the etched aluminium surface. 

Both modified aluminium surfaces provide highly efficient corrosion-resistant properties during immersion in 0.1 M NaCl solution. The FAS-8-treated surface remains unchanged even after 2 weeks of exposure in the salt spray chamber according to standard ASTM B117-19. Moreover, aluminium grafted with 1H,1H,2H,2H-perfluorooctyltrimethoxysilane indicates a self-cleaning ability and melting delay. 

The presented etching concept can be further used to fabricate large-scale superhydrophobic aluminium surfaces used for various industrial applications.

## Figures and Tables

**Figure 1 molecules-27-01099-f001:**
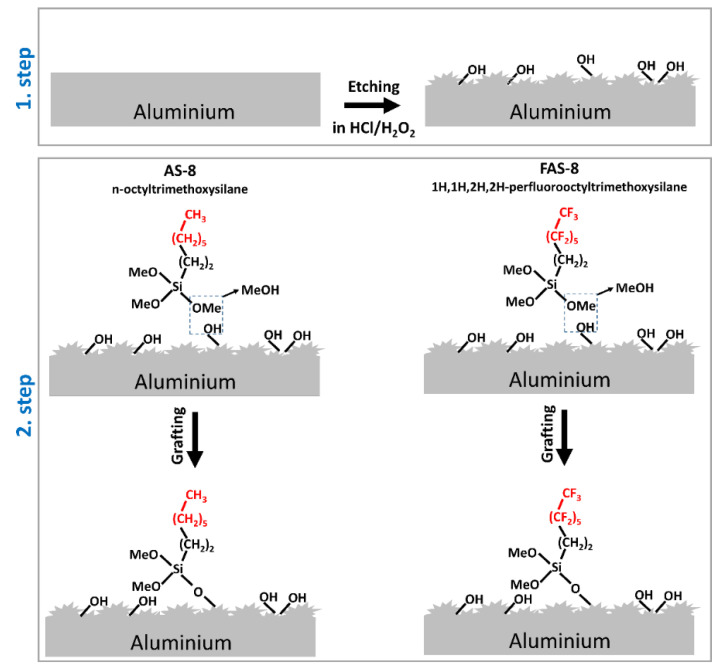
Schematic illustration of the formation of a (super)hydrophobic aluminium surface prepared by the etching process in the HCl/H_2_O_2_ solution (1st step) followed by grafting with n-octyltrimethoxysilane (AS-8) and 1H,1H,2H,2H-perfluorooctyltrimethoxysilane (FAS-8) (2nd step).

**Figure 2 molecules-27-01099-f002:**
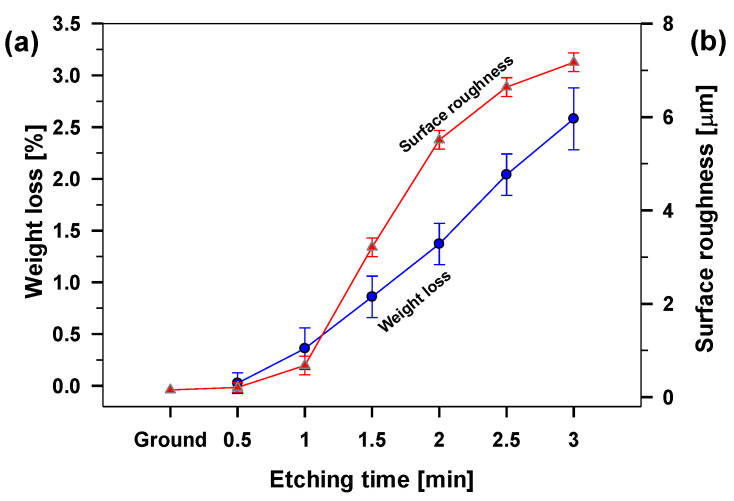
(**a**) weight loss of aluminium during the etching process after selected immersion times (from 0.5 to 3 min) in the HCl/H_2_O_2_ solution and (**b**) their respective surface roughness (*S*_a_). A solid connecting line is drawn between the obtained data points.

**Figure 3 molecules-27-01099-f003:**
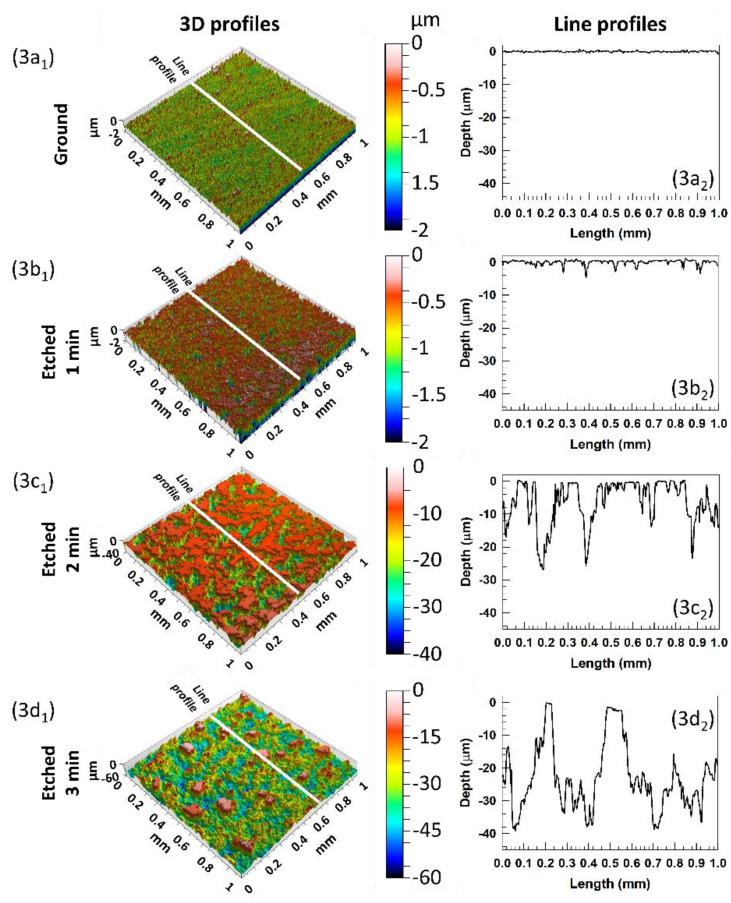
3D surface topography images of ground aluminium surface and etched for 1, 2 and 3 min in HCl/H_2_O_2_ solution (left panel). The white lines present the area where single line profiles analyses were performed, and their corresponding line profiles are presented in right panel. The surface roughness determined from 3D profiles is presented in Figure 2b.

**Figure 4 molecules-27-01099-f004:**
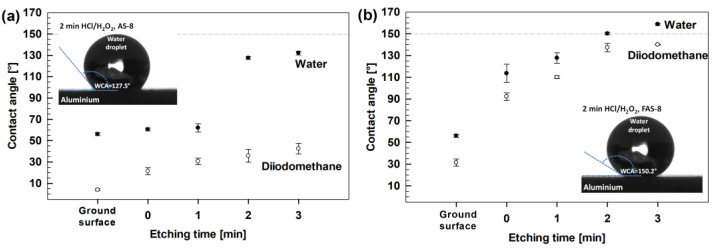
Water and diiodomethane contact angles measured of aluminium surface etched for selected times in HCl/H_2_O_2_ and treated for selected times min in 1 wt.% (**a**) AS-8 and (**b**) FAS-8 solutions. The results are presented as mean value ± standard deviation. The dashed line presents the boundary of the superhydrophobic surface (WCA > 150°). The inset images in Figure 4 presents the water drop on 2 min etched aluminium treated with AS-8 and FAS-8 film.

**Figure 5 molecules-27-01099-f005:**
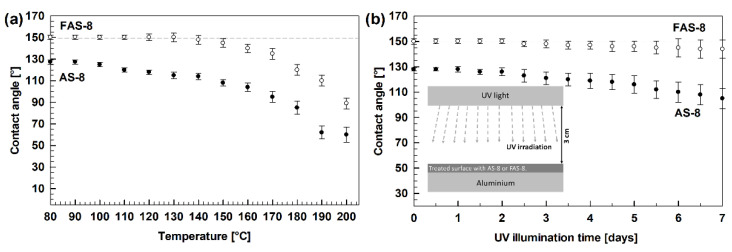
Water contact angles during (**a**) thermal stability testing during annealing for one hour at each selected temperatures and (**b**) UV durability during irradiation exposure. The insert illustration in (**b**) presents the set-up of the UV test.

**Figure 6 molecules-27-01099-f006:**
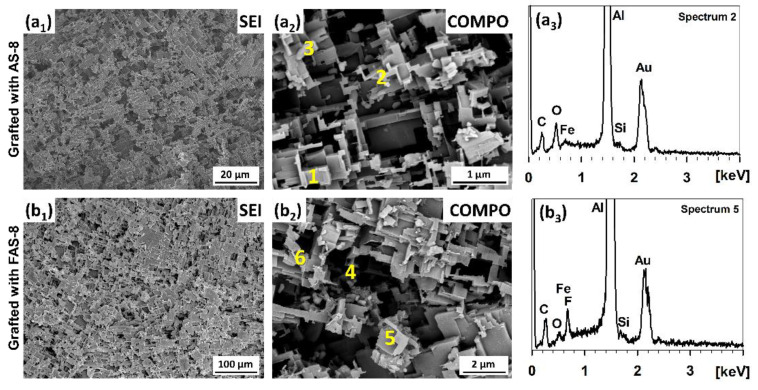
SEM images of the aluminium surface etched for 20 min in 1 M HCl/H_2_O_2_ treated for 30 min in 1 wt. % AS-8 and FAS-8 solutions recorded in SEI and COMPO modes. The enumerated positions (from 1 to 6) indicate spots where the EDS analyses were performed. Values are given in Table 1; EDS spectra 2 and 5 are shown as representatives.

**Figure 7 molecules-27-01099-f007:**
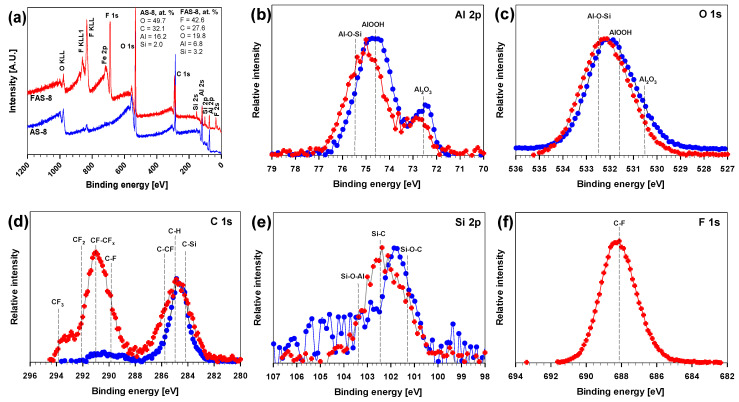
(**a**) survey XPS spectra with the inserted composition of AS-8- and FAS-8-treated aluminium surfaces given in atomic percentage (at. %) and high-resolution XPS spectra (**b**) Al 2p, (**c**) O 1s, (**d**) C 1s, (**e**) Si 2p and (**f**) F 1s spectra for aluminium surface etched for 2 min in HCl/H_2_O_2_ followed by grafting for 30 min in 1 wt.% AS-8 and FAS-8 solutions.

**Figure 8 molecules-27-01099-f008:**
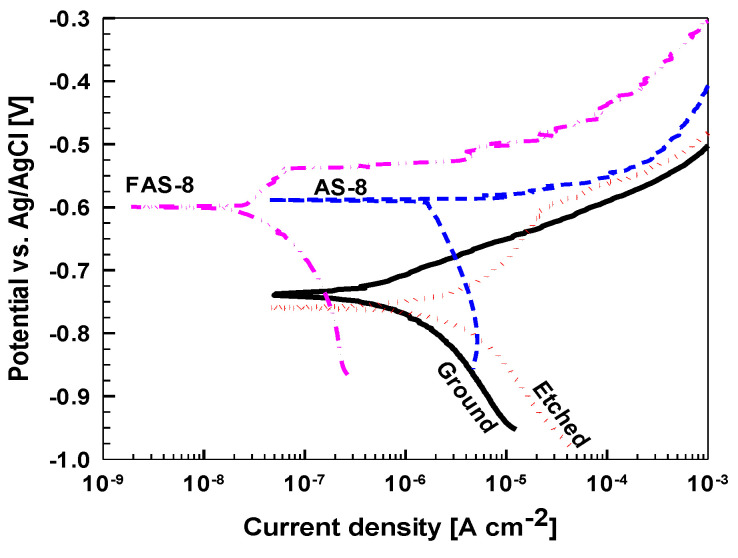
Potentiodynamic polarization curves for ground, etched and treated aluminium with AS-8 and FAS-8 measured in 0.1 M NaCl after immersion for one hour. d*E*/d*t* = 1 mV/s.

**Figure 9 molecules-27-01099-f009:**
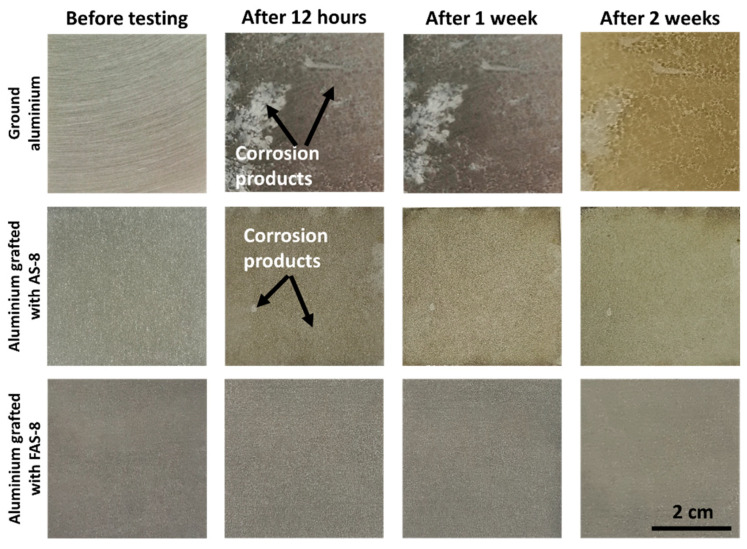
The surface appearance of ground aluminium and treated aluminium surface with AS-8 and FAS-8 after selected exposure times in the salt spray chamber, test performed according to standard ASTM B117-19, i.e., 5 % NaCl, 35 °C.

**Figure 10 molecules-27-01099-f010:**
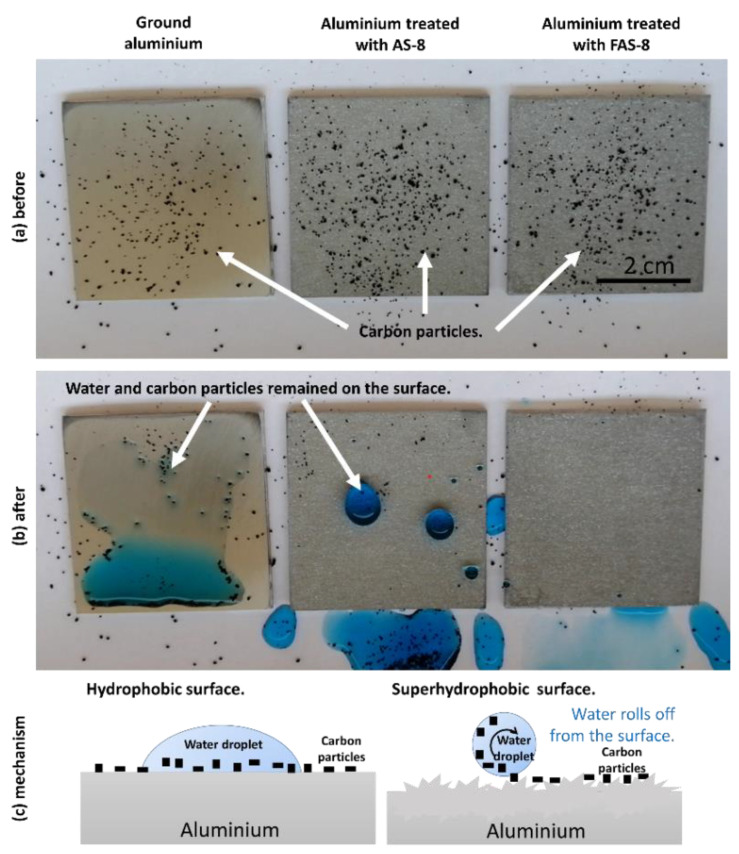
The surface appearance of ground and treated aluminium with an AS-8 and treated aluminium with a FAS-8 covered with carbon particles (**a**) before and (**b**) after rinsing with tap water. Figure (**c**) presents a self-cleaning mechanism on a hydrophobic and superhydrophobic surface.

**Figure 11 molecules-27-01099-f011:**
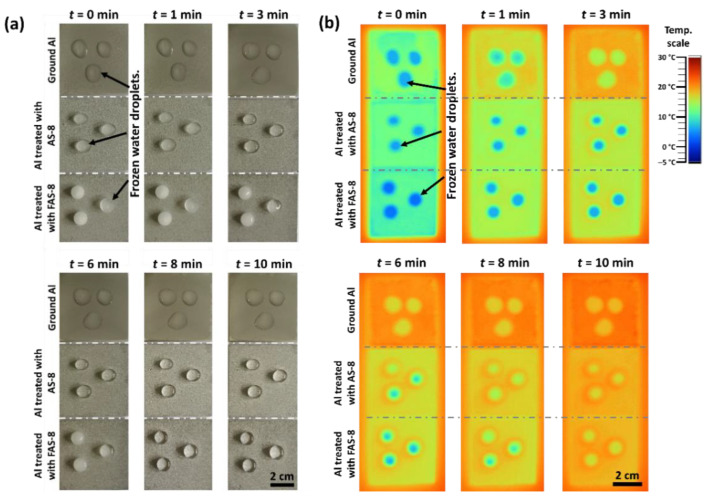
(**a**) testing the anti-icing properties of ground and treated aluminium with AS-8 and FAS-8 recorded at selected times after taking samples out from the freezer (from –15 °C) and leaving them at ambient temperature. (**b**) was recorded using a thermal infrared camera.

**Table 1 molecules-27-01099-t001:** EDS analysis of treated aluminium with AS-8 and FAS-8 given in weight percentage (wt. %) on the numbered spots in Figure 6.

Scheme	Al	O	Fe	C	Si	F
1	92.4	1.9	-	5.3	0.4	-
2 (spectrum 2)	68.8	8.3	1.7	20.5	0.7	-
3	54.9	4.7	27.9	9.3	3.2	-
4	84.9	2.3	-	12.0	0.1	0.7
5 (spectrum 5)	81.8	1.3	10.2	5.1	0.5	1.1
6	63.3	7.0	11.0	15.1	0.8	2.8

**Table 2 molecules-27-01099-t002:** Evaluated electrochemical parameters with standard deviation in 0.1 M NaCl for ground, etched and treated aluminium with AS-8 and FAS-8: corrosion current density (*j*_corr_), corrosion potential (*E*_corr_) and pitting potential (*E*_pit_) were determined from potentiodynamic polarization curves (Δ*E* = |*E*_pit_ − *E*_corr_|) in Figure 8.

	*j*_corr_ [µA/cm^2^]	*E*_corr_ [V]	*E*_pit_ [V]	Δ*E* [mV]
Ground	0.51 ± 0.05	−0.73 ± 0.02	−0.73 ± 0.01	10
Etched	2.38 ± 0.10	−0.76 ± 0.02	−0.60 ± 0.01	160
Treated using AS-8	1.72 ± 0.08	−0.59 ± 0.03	−0.58 ± 0.01	10
Treated using FAS-8	0.03 ± 0.002	−0.60 ± 0.03	−0.54 ± 0.01	60

## Data Availability

The data presented in this study are all given in the present article.

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
