# Peer review of "Superhydrophobic Aluminium Surface to Enhance Corrosion Resistance and Obtain Self-Cleaning and Anti-Icing Ability"

_molecules, 2022, doi:10.3390/molecules27031099_

Round 1
Reviewer 1 Report
Comments to the authors are attached below as PDF document.

Author Response
In this article, the authors describe a two-step facile fabrication technique to fabricate superhydrophobic aluminium surface to enhance corrosion resistance and obtain self-cleaning and anti-icing ability. The manuscript is well written with organised high-quality data. The topography, chemical and wetting characterisation of the fabricated surface were done with various techniques. Corrosion tests were rigorous and the substrate shows good anti corrosion properties with high stability against harsh conditions such as high temperature and UV exposure. In addition, authors demonstrate the self-cleaning and anti-icing ability of the superhydrophobic aluminium substrates. Therefore, I suggest to accept this manuscript after some minor corrections.
Dear Reviewer, thank you for reviewing the manuscript and writing a constructive, positive opinion. The replies to your comments are below, and the manuscript is corrected following your suggestions. The changes in the original manuscript text are yellow shaded.
Answers to all your and other reviewers` comments are given in the attached file.
Comment 1: Page 7, 3. Results and discussion, Line 292 “The etching rate of aluminium in mixture HCl/H2O2 was evaluated from the shape of 292 the curve obtained from weight loss calculations after selected times”. Here “shape” should be replaced by the word “slope”.
Thank you very much for the comment. The word “shape” was replaced by the word “slope”.
Comment 2: In wetting characterisation section, as a test liquid diiodomethane was used other than water. Why particularly diiodomethane was used?
The wettability of the surface is usually measured with water and additionally with some other (less)polar or non-polar liquids (diiodomethane, formamide, i.e. ref: https://doi.org/10.1016/j.apsusc.2017.01.068). In the current research, the diiodomethane was used to demonstrate the difference between AS-8 and FAS-8 molecules (effect of CH3 and CF3 chain in the AS-8 and FAS-8 molecule). The difference in the contact angles is not just the effect of grafting, but it is also related to to the nature of the alkyl chain.
What is the surface tension of the diiodomethane? It should be mentioned in the manuscript that will help the reader to correlate the understanding of the apparent contact angle of the diiodomethane on superhydrophobic surface.
γl is the polar surface energy for the test liquids (diiodomethane γl = 50.8 mN/m, water γl = 72.8 mN/m), i.e. ref: dx.doi.org/10.1021/la5018328. We now included this information in the manuscript (added new reference 59).
It is also clear from the wetting studies that the fabricated surface is not superoleophobic as it does not posses the re-entrant roughness on the substrate. An optical image of the Diiodomethane drop with contact angle should also be put in the inset.
The manuscript confirmed superhydrophobicity for FAS-8 treated surface, which was the main aim. We agree that additional testing calculations of surface energy can be done, but this will be the topic for further research.
We think that the images of diiodomethane drop do not give any additional information.
Figure 1. The images present the diiodomethane drop on 2-min etched aluminium treated with AS-8 (left) and FAS-8 (right) coatings.
Comment 3: In the durability test with high temperature, heating time should be mentioned?
This information is already explained in the experimental part, Section 2.2.4. “The thermal stability of the modified aluminium surface was checked by annealing for one hour the treated aluminium at various temperatures from 80 to 200 °C at an interval of 10 °C.”
This information is now was also given in the figure description (Fig. 5).
Comment 4: Page 14, 3.6.1. Potentiodynamic polarisation measurements , Line 544 “ground Al < FAS-8 < AS-8” Here it should be “ground Al < AS-8 < FAS-8”.
Thank you very much for the comment. We apologise for this mistake. The text in the manuscript is corrected to Al < AS-8 < FAS-8.

Reviewer 2 Report
The authors fabricated superhydrophobic surface of aluminium by chemical etching and investigated its several applications, including anti-corrosion and anti-icing.
- The authors shall discuss the reasons for the variation of hydrophilicity and hydrophobicity with the surface roughness, with and without the low SE treatment.
- Are there any approaches to enhance the thermostability further?
- Section 3.3 could be moved after sections 3.4. and 3.5
- Providing references to a few published reviews could be helpful to the readers. For example, for the sentence “Up to now, various artificial superhydrophobic surfaces have been fabricated by tailoring surface topography and using techniques such as electrodeposition [30,44], anodic oxidation [45],…...”, the authors shall provide more comprehensive reviews as references. Several reviews are available on electrodeposition ( https://doi.org/10.1002/asia.202001425) or anodic oxidation (https://doi.org/10.1016/j.cis.2020.102245) or other methods.
Author Response
Dear Reviewer, thank you for reviewing the manuscript and writing a constructive, positive opinion. The replies to your comments are below, and the manuscript is corrected following your suggestions. The changes in the original manuscript text are yellow shaded.
Answers to all your and other reviewers` comments are given in the attached file.
- The authors shall discuss the reasons for the variation of hydrophilicity and hydrophobicity with the surface roughness, with and without the low SE treatment.
The reason for hydrophilicity is the etching process in HCl/H2O2 solution, which removes the native passive film. After etching under such acidic conditions, the newly formed aluminium oxide film is formed consisting of aluminium oxide/hydroxide (see Figure 1). The presence of these hydroxyl groups on the surface makes the surface hydrophilic. This was explained in the manuscript in section 3.1. and also in the Introduction, 3rd paragraph.
“Prepared mixture HCl/H2O2 solution has pH ~ 1, under which conditions the aluminium oxide passive layer is not resistant, and aluminium reacts vigorously. The summary of the chemical process on the aluminium surface during etching is presented with Equations 3-5. Aluminium reacts with acid (HCl), forming AlCl3. The side product of this reaction is H2. At the same time, formed AlCl3 spontaneously reacts with hydrogen peroxide forming the aluminium hydroxide, Al(OH)3.”
- Are there any approaches to enhance the thermostability further?
Thank you very much for your questions. Yes, these are options, such as increasing the film thickness with a longer immersion time (i.e. 1 day). There are also some other approaches, such as electrochemical deposition, which allows the formation of a denser and thicker film on the surface. However, in this study, we focused on forming very thin films during the immersion process.
- Section 3.3 could be moved after sections 3.4. and 3.5
We prefer to leave section 3.3 at the same position in the manuscript because it follows section 3.2, where the results related to water wettability was described.
- Providing references to a few published reviews could be helpful to the readers. For example, for the sentence “Up to now, various artificial superhydrophobic surfaces have been fabricated by tailoring surface topography and using techniques such as electrodeposition [30,44], anodic oxidation [45],…...”, the authors shall provide more comprehensive reviews as references. Several reviews are available on electrodeposition ( https://doi.org/10.1002/asia.202001425) or anodic oxidation (https://doi.org/10.1016/j.cis.2020.102245) or other methods.
Thank you very much for your suggestions. We include these suggested references in the manuscript (added new references 45 and 47).

Reviewer 3 Report
1. I strongly recommend for revising the manuscript for grammatical and typing errors.
2. For further improvement in the corrosion part of the manuscript it is suggested to add the following references in the manuscript
a. Fundamentals of Materials Engineering-A Basic Guide, Bentham Science publishers, ISBN: 978-981-14-8920-4, DOI: 10.2174/97898114892281210101.
b. Arch. Metall. Mater. 63 (2018), 2, 749-763
c. Anal. Bioanal. Electrochem., Vol. 10, No. 3, 2018, 349-361
3. Authors has written “The aluminium samples were immersed into freshly prepared etching solution in the glass tempering beaker to control etching solution temperature between 25 and 40 °C”. It is recommended to add the duration of the etching.
4. Is there any particular reason to dry the grafted samples using a stream of compressed nitrogen? Because there is a chance that nitrogen can react with the sample.
5. For treated using AS-8 samples there is an abrupt increase in current density. Why?
6. Fig. 9 seems optical micrographs and I request authors to increase the magnification of the scale for the better observation of the surface of the materials.
7. In Fig. 11, it seems at zero minute (I mean immediately after placing ice on the material) itself ice is melted, how is it possible?

Author Response
Dear Reviewer, Thank you for reviewing the manuscript and writing a constructive, positive opinion. The replies to your comments are below, and the manuscript is corrected following your suggestions. The changes in the original manuscript text are yellow shaded.
Answers to all your and other reviewers` comments are given in the attached file.
- I strongly recommend for revising the manuscript for grammatical and typing errors.
Thank you very much for your opinion. The manuscript will be sent for grammatical and typing errors to the MDPI service (it is the next step in the submission process).
- For further improvement in the corrosion part of the manuscript it is suggested to add the following references in the manuscript
- Fundamentals of Materials Engineering-A Basic Guide, Bentham Science publishers, ISBN: 978-981-14-8920-4, DOI: 10.2174/97898114892281210101.
- Arch. Metall. Mater. 63 (2018), 2, 749-763
- Anal. Bioanal. Electrochem., Vol. 10, No. 3, 2018, 349-361
Thank you very much for your suggestion. The manuscript was submitted to the journal Molecules (special issue: Superhydrophobic and Superoleophobic Materials; https://www.mdpi.com/journal/molecules/special_issues/superhydrophobic_superoleophobic_materials), which idea was presenting the developments of superhydrophobic/superoleophobic surfaces, mainly focusing on their design principles, fabrication methods, and less on presenting specific corrosion characterisation using various electrochemical methods. Therefore we presented only results obtained with potentiodynamic measurements to confirm enhanced corrosion properties after surface treatment with AS-8 and FAS-8.
We checked the suggested references, but unfortunately, they are not directly related to the manuscript’s topic because they describe the effect of Y2O3 nanoparticles or sintering temperature on stainless steel corrosion using the linear sweep voltammetric method. However, thank you very much for a nice idea of how linear sweep voltammetric methods can be used for further corrosion characterisation.
- Authors has written “The aluminium samples were immersed into freshly prepared etching solution in the glass tempering beaker to control etching solution temperature between 25 and 40 °C”. It is recommended to add the duration of the etching.
The etching time was different, ranging from 0.5 to 3 min (see Figure 2). Most of the results in sections 3.3 – 3.8 are presented for 2 minutes etched aluminium (where superhydrophobic properties was obtained for aluminium treated with FAS-8). However, we modified the text to clarify this parameter (page 3, line 123).
- Is there any particular reason to dry the grafted samples using a stream of compressed nitrogen? Because there is a chance that nitrogen can react with the sample.
The samples were dried with nitrogen to dry the surface after the etching and grafting process. If the surface is dried with compressed air, the surface can be contaminated with pump oil or some other impurities. However, the nitrogen is inert to aluminium, so no reaction is expected with the freshly etched aluminium surface.
- For treated using AS-8 samples there is an abrupt increase in current density. Why?
The aluminium treated with AS-8 decreased the corrosion current density compared to the etched surface from 2.38 mA/cm2 to 1.72 mA/cm2, Figure 8, Table 2. Despite that, however, once passing the corrosion potential, the current density increases almost immediately, reflecting the inability of this film to prevent the pitting corrosion.
- Fig. 9 seems optical micrographs and I request authors to increase the magnification of the scale for the better observation of the surface of the materials.
Thank you for the comment. The corrosion process during standard test ASTM B117 is usually evaluated with the naked eye. Therefore the figures at higher magnification are usually not needed. However, to see the corrosion products more clearly, we tried to enhance the quality of the images (increase the figure sharpness), especially for ground and AS-8 treated surfaces, Figure 9.
- In Fig. 11, it seems at zero minute (I mean immediately after placing ice on the material) itself ice is melted, how is it possible?
Thank you for your comment. The ice was not melted immediately after placing ice on the material. This can be seen in Fig. 11b, where the image confirmed that the ice was still present after 1 min. It was melted after 3 minutes. This melting delay is in accordance with thermodynamics. The droplet on the cold surface gains heat from the air in the forms of heat conduction and thermal radiation and absorbs heat by heat conduction, reference 37.

Reviewer 4 Report
General comments
In the present study, superhydrophobic aluminium surface was fabricated by chemical etching with HCl/H2O2 and chemical grafting with n-octyltrimethoxysilane (AS-8) and 1H,1H,2H,2H-perfluo-rooctyltrimethoxysilane (FAS-8). The self-cleaning and anti-icing performances of the fabricated aluminum surface were studied. The manufactured surface showed superhydrophobicity and self-cleaning property. After read through the whole manuscript, several concerns are given below:
- The authors reported the fabrication of aluminum surface and characterized the superhydrophobicity, self-cleaning property and anti-icing performances. However, the study on the anti-icing performance is not profound enough. For instance, the freezing time of water droplet on the surface should be studied, the effects of surface structure and surface chemistry on the anti-icing behavior should be investigated. Besides, the whole manuscript looks like recording the experiment results and lack of sufficient discussion and analysis.
- Will the surface morphology after etching affect the anti-icing? How did the authors precisely control the surface structure? What is the etching concentration of HCl/H2O2 solution?
- How to precisely control the chemical grafting?
- The author mentioned “After 1 hour, the specimens were taken out and left at ambient temperature” in the section of 2.2.9 anti-icing, what is the value of ambient temperature?
- It is better to provide a diagram to describe the etching process.
- What is the novelty of this study? What is the research focus of this study?
- The Introduction is not well organized and failed to illustrate the motivation of this work.
Though the manuscript reported one method to develop superhydrophobic, self-cleaning surface and studied their anti-corrosion and anti-icing behavior, the motivation of this work is not very clear and lack of sufficient novelty. In the current stage, major revision should be given before it can be considered for publication on Molecules.
Author Response
In the present study, superhydrophobic aluminium surface was fabricated by chemical etching with HCl/H2O2 and chemical grafting with n-octyltrimethoxysilane (AS-8) and 1H,1H,2H,2H-perfluo-rooctyltrimethoxysilane (FAS-8). The self-cleaning and anti-icing performances of the fabricated aluminum surface were studied. The manufactured surface showed superhydrophobicity and self-cleaning property. After read through the whole manuscript, several concerns are given below:
- The authors reported the fabrication of aluminum surface and characterised the superhydrophobicity, self-cleaning property and anti-icing performances. However, the study on the anti-icing performance is not profound enough. For instance, the freezing time of water droplet on the surface should be studied, the effects of surface structure and surface chemistry on the anti-icing behavior should be investigated. Besides, the whole manuscript looks like recording the experiment results and lack of sufficient discussion and analysis.
Dear Reviewer, Thank you for reviewing the manuscript and writing a constructive, positive opinion. The replies to your comments are below, and the manuscript is corrected following your suggestions. The changes in the original manuscript text are yellow shaded.
Answers to all your and other reviewers` comments are given in the attached file.
- Will the surface morphology after etching affect the anti-icing?
The etched surface is super hydrophilic, meaning the water completely wets the surface (water contact angles are very low). Consequently, the ice is in contact with the surface and ice is strongly adhered to the surface, opposite to the treated surfaces. For that reason we did not present the data on non-treated, etched Al surface, and only treated surfaces were compared with ground aluminium surface.
How did the authors precisely control the surface structure?
The surface structure was controlled with etching time, presented in Fig. 3.
What is the etching concentration of HCl/H2O2 solution?
The preparation and concentration of the etching solution is described in Experimental section 2.1.
“The etchant solution was prepared by mixing H2O (Milli-Q Direct water with the resistivity of 18.2 MΩ cm at 25 °C, Millipore, Billerica, MA), 37 wt. % HCl (CAS No. 7647-01-0, ACS reagent, Sigma-Aldrich, Germany), and 30 wt. % H2O2 CAS No. 7722-84-1, ACS reagent, Sigma-Aldrich, Germany) in the volume ratio of 10 : 2: 3.”
- How to precisely control the chemical grafting?
The chemical grafting is also described in section 2.1.
“The grafting of the etched samples was performed by immersion in freshly prepared 1 wt. % ethanol solution containing n-octyltrimethoxysilane (AS-8, CAS No. 3069-40-7) or 1H,1H,2H,2H-perfluorooctyltrimethoxysilane (FAS-8, CAS No. 85857-16-5), both dis-tributed from Fluorochem, United Kingdom, Europe and used without any purification. The samples were placed in a beaker with the etched surface facing up. This process was performed at ambient temperature. After 30 minutes, the samples were taken out, thoroughly rinsed with distilled water and ethanol, and dried with a stream of compressed nitrogen.”
- The author mentioned “After 1 hour, the specimens were taken out and left at ambient temperature” in the section of 2.2.9 anti-icing, what is the value of ambient temperature?
The ambient temperature meant room temperature, between 21-24 °C.
- It is better to provide a diagram to describe the etching process.
The etching process is described with equations 3-5, so we think that this is demonstrative enough to understand the etching process of aluminium in the HCl/H2O2 solution.
- What is the novelty of this study? What is the research focus of this study?
The manuscript was submitted to the journal Molecules (special issue: Superhydrophobic and Superoleophobic Materials; https://www.mdpi.com/journal/molecules/special_issues/superhydrophobic_superoleophobic_materials), which was devoted to presenting the ideas of developing superhydrophobic/superoleophobic surfaces, mainly focusing on their design principles fabrication methods.
However, the main novelties in this manuscript are:
- facile (immersion in etching solution) and fast (only few minutes) etching process of aluminium
- facile and fast (only 30 minutes) grafting the etched surface during immersion in solution containing AS-8 or FAS-8
- grafting the surface with fluoro(alkyl) methoxy silanes and compared their behaviour with alkyl antipode
- anti-icing properties were evaluated with the infrared camera.
- The Introduction is not well organised and failed to illustrate the motivation of this work.
Thank you very much. We tried to describe in the Introduction the need for the development of superhydrophobic surfaces. the state-of-the-art of the procedures used and also presented some general considerations of hydrophilic and hydrophobic surfaces. We summarized our previous results and pointed out what was the aim of the present study. In our opinion, the text contains the most important points one should present in the Introduction and is in accordance with the Instructions for Authors (https://www.mdpi.com/journal/molecules/instructions). It is a pity that the Reviewer did not describe in more detail what it means “not well organized”. We now added several new references related to the commonly used techniques to fabricate superhydrophobic surfaces.
Though the manuscript reported one method to develop superhydrophobic, self-cleaning surface and studied their anti-corrosion and anti-icing behavior, the motivation of this work is not very clear and lack of sufficient novelty. In the current stage, major revision should be given before it can be considered for publication on Molecules.
We hope that the replies given and the changes introduced will convince the Reviewer that the manuscript brings sufficiently novel data to be published.

Round 2
Reviewer 4 Report
The authors have addressed all of the questions, the manuscript can be accepted.